

# Exploitation of the far-offshore wind energy resource by fleets of energy ships. Part B. Cost of energy

Aurélien Babarit[1], Simon Delvoye[1], Gaël Clodic[1], Jean-Christophe Gilloteaux[1]

[1]LHEEA, Ecole Centrale de Nantes - CNRS, Nantes, 44300, France

*Correspondence to*: Aurélien Babarit (aurelien.babarit@ec-nantes.fr)

**Abstract.** This paper deals with a new concept for the conversion of far-offshore wind energy into sustainable fuel. It relies on autonomous sailing energy ships and manned support tankers. Energy ships are wind-propelled ships that generate electricity using water turbines attached underneath their hull. Since energy ships are not grid-connected, they include onboard power-to-X plants for storage of the produced energy. In the present work, the energy vector is methanol.

In the first part of this study (Babarit et al., submitted), an energy ship design has been proposed and its energy performance has been assessed. In this second part, the aim is to estimate the energy and economic performance of the whole system. Thus, an energy and economic model has been developed which is presented in the paper. Results show that an initial FARWIND system could produce approximately 100,000 tonnes of methanol per annum (approximately 550 GWh per annum of chemical energy) at a cost in the range 150 to 325 €/MWh, and that FARWIND-produced methanol could compete

with gasoline on the EU transportation fuel market in the long term.

## 1 Introduction

To date, fuels such as oil, natural gas and coal account for approximately 80% of primary energy consumption globally (BP, 2018). Although this share is expected to decrease with the development of renewable power generation and the electrification of the global economy, some sectors may be difficult to electrify (e.g. aviation, freight). Therefore, if a global

temperature change of less than 2°C—as set out in the Paris agreement—is to be achieved, there is a critical need to develop low-carbon alternatives to fossil fuels.

To address this challenge, we proposed in (Babarit et al., submitted) an energy system (FARWIND) which could convert the far-offshore wind energy resource into sustainable fuel using fleets of energy ships, see Fig. 1. Energy ships are ships propelled by the wind which generate electricity by means of water turbines attached underneath their hulls. The generated

electricity is converted into fuel using onboard power-to-gas (PtG) or power-to-liquid (PtL) plants. In the proposed system, the fuel is methanol. It is collected by tankers which are also used to supply the energy ships with the necessary feedstock (carbon dioxide) for power-to-methanol conversion. Of course, the $CO_2$ supply source must be sustainable for that system to produce sustainable methanol. Therefore, the $CO_2$ must be captured either directly or indirectly from the atmosphere.





Possible options include direct air capture (Keith et al., 2018), $CO_2$ capture from flue gases from biomass or FARWIND-produced methanol combustion, and CO2 from biogas upgrading (Li et al., 2017; Irlam, 2017).

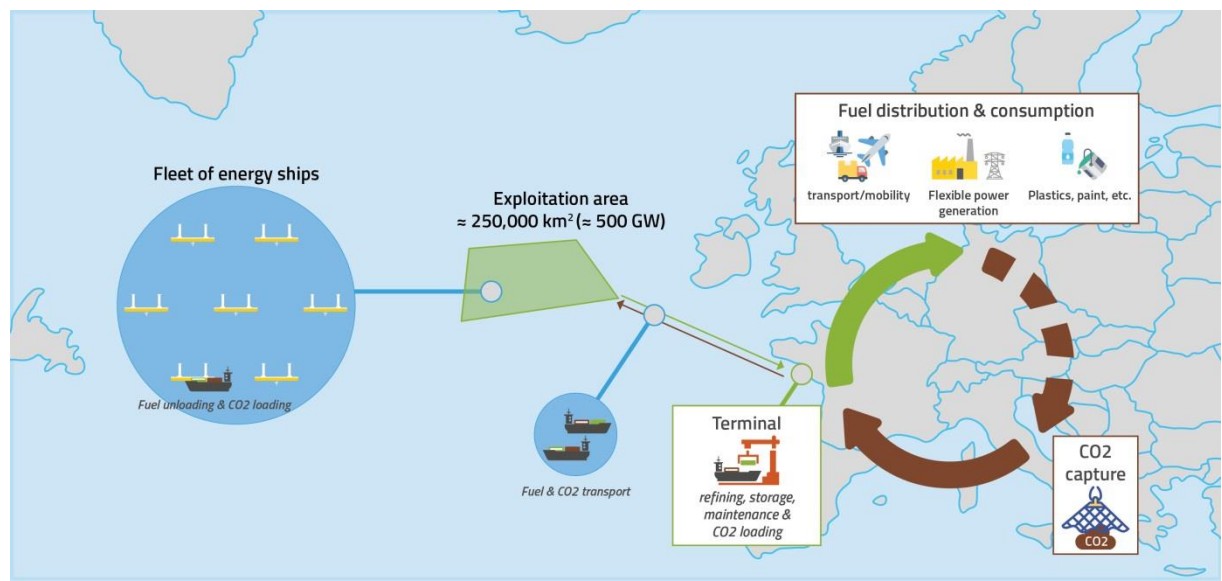

**Figure 1 The concept of sustainable methanol production from far-offshore wind energy by FARWIND energy systems.**

The overall aim of the present study is to investigate the energy and economic performance of the FARWIND energy system. An energy ship design was proposed in a previous paper (Babarit et al., submitted) and its energy performance was investigated. Elaborating on these results, the aim of the present paper is to estimate the associated cost of energy.

The remaining of this paper is organized as follows. In section 2, the specifications of the proposed energy system are presented and its annual methanol production is estimated. Estimates of expenditures for the proposed energy system are provided and discussed in section 3. Using those estimates and the estimates of annual methanol production, the cost of energy is estimated in section 4 and market perspectives are discussed. Conclusions are presented in Section 5.



## 2 Specifications of the proposed FARWIND system

### 2.1 Energy ship design

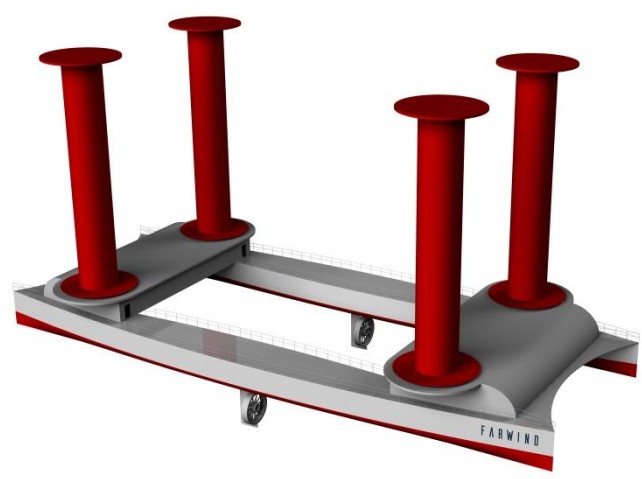

**Figure 2  Artist's view of the proposed energy ship design.**

The energy ship design considered in this study is that presented in (Babarit et al., submitted). It consists of an 80 m long catamaran with four 30 m tall Flettner rotors, and two water turbines, each at rated power 900 kW each. Fig. 2 shows an artist impression of the proposed design. Its characteristics are recalled in Tab. 1. According to (Babarit et al., submitted), its capacity factor is expected to be 75% corresponding to an annual methanol production of 905 tonnes.

|  | Unit | Value |
|---|---|---|
| Hull |  |  |
| Length | m | 80 |
| Breadth | m | 31.7 |
| Draught | m | 1.6 |
| Displacement | t | 660 |
| Structural mass | t | 258 |
| Wind propulsion |  |  |
| Type | - | Flettner rotors |
| Number | - | 4 |
| Rotor height | m | 30 |
| Rotor diameter | m | 5 |
| Rotor mass | t | 59 |
| Rotor rated power | kW | 110 |
| Water turbine |  |  |
| Number | - | 2 |
| Turbine diameter | m | 4 |
| Rotor-to-electricity efficiency ($\eta_3$) | - | 80% |

| Turbine mass | t | 7.4 |
|---|---|---|
| Rated power | kW | 900 |
| Auxiliaries subsystems | | |
| Power consumption | kW | 50 |
| Auxiliaries subsystems mass | t | 32 |
| Power-to-methanol plant | | |
| Electrolyzer rated power | kW | 1,420 |
| Electrolyzer mass | t | 35 |
| Desalination unit rated power | kW | Negligible |
| Desalination unit mass | t | Negligible |
| $H_2$tMeOH plant capacity | kg/h | 138 |
| H2tMeOH plant mass | t | 24 |
| Storage tanks | | |
| $CO_2$ storage capacity | t | 32 |
| Storage tank mass (empty) | t | 21 |
| Methanol storage capacity | t | 23 |
| Storage tank mass | t | 5 |

**Table 1 Specifications of the proposed energy ship design**

## 2.2 Tanker design

In the FARWIND concept, the energy ships are deployed in fleets and are supported by tankers which collect the produced methanol and transport it to a shore-based terminal. The tankers also provide the energy ships with $CO_2$.

In the considered energy ship design, the methanol storage tank capacity allows storage of one week of methanol production. Therefore, each and every energy ship of the fleet must meet a tanker for methanol collection and $CO_2$ refill at least once a week (to avoid stops in the production process because the methanol tank is full or because the $CO_2$ tank is empty).

Thus, let us estimate the number of energy ships that can be served by one tanker. This depends on the duration of the $CO_2$-loading and methanol-unloading operations. We assume that these operations take six hours on average, and that they are carried out continuously (including at night). Therefore, one tanker can service 28 energy ships per week (7 days/week x 24 hours/day / 6 hours/operation). As the capacity of an energy ship's methanol tank is 23 tonnes (32 tonnes for the $CO_2$ tank), the tanker may collect up to 644 t of methanol and supply 896 t of $CO_2$ every week.

It is assumed that the tankers are operated by a crew, and that the duration of their mission is four weeks. At the end of each four-week mission, the tanker returns to a shore-based terminal to change crew, unload the methanol and load $CO_2$.





Therefore, their total methanol capacity must be 2,576 t (4 weeks x 644 t/week) and their total $CO_2$ capacity must be 3,584 t (4 w x 896 t/w). Assuming the $CO_2$ will be stored as liquid in a cryogenic storage tank, and extrapolating from (Chart, 2019),
the empty weight of a 3,584 t capacity $CO_2$ storage vessel is estimated to be 2,240 t. For methanol, the mass of the required tank is estimated to be 518 t. The tanker will be carrying maximum cargo weight (6,342 t) when it leaves the terminal (full $CO_2$ tank and empty methanol tank). This cargo weight is relatively similar to the average vessel size of small crude oil (3,600 deadweight (dwt)), chemical (4,900 dwt) and LPG vessels (3,500 dwt) (Lindstad et al., 2012). According to (MAN Energy Solutions, 2019), the propulsion power of a 5,000 t deadweight bulk carrier is 1,410 kW for a service speed of 12
knots. These are the values used for the service speed and propulsion power of the tanker.

## 2.3 FARWIND system design and annual methanol production

Following (Babarit et al., 2018), it is assumed that the fleet of energy ships is deployed at a distance of 1,000 km from the terminal. Therefore, the tankers have to travel 1,000 km to meet the energy ships, and a further 1,000 km when returning to the terminal. At a service speed of 12 knots, the tanker's round-trip will take 90 hours. Taking into account the duration of
unloading/loading operations and other maintenance operations, we estimate that the tanker will be away from the fleet of energy ships for a duration of one week.

To ensure continuous operation of the energy ships, the tanker must be replaced immediately when it leaves the production zone. Therefore, each group of 28 energy ships must be supported by more than one tanker. It can be shown that the minimum number of tankers per fleet must be at least 1.25, meaning that the optimal FARWIND system comprises a fleet of
112 energy ships supported by five tankers. Over a year, the number of round-trips between the terminal and the production zone is 10.4 for each tanker. The maximum methanol production of that system (assuming 100% capacity factor for the energy ships) is approximately 135,000 t per annum. Assuming an average capacity factor of 75%, the annual methanol production would be approximately 100,000 t per annum. Note that it would require the supply of approximately 140,000 t of $CO_2$.

## 3 Estimation of expenditures

### 3.1 Capital cost of a first of a kind energy ship

|  | Cost (k€) |
|---|---|
| **Energy ship** |  |
| Hull | 1,290 – 5,160 k€ |
| Flettner rotors | 3,000 – 3,500 k€ |
| Water turbine | 488 k€ |
| Electrolyzer | 1,280 – 1,700 k€ |
| $H_2$-to-methanol plant | 1,030 – 1,380 k€ |



| | |
|---|---|
| Fresh water production unit | 11 k€ |
| Liquid $CO_2$ tank | 32 k€ |
| Methanol tank | 7 k€ |
| Auxiliaries, assembly and systems integration | 1,430 – 2,450 k€ |
| **Total** | **8,570 – 14,700 k€** |

**Table 2 Estimates of the capital cost of a prototype of the proposed energy ship**

Tab. 2 shows estimates of the capital cost of a prototype of the proposed energy ship. The water turbine cost and the fresh water production unit cost were estimated using scaling laws which were developed by Holl et al. (Holl et al., 2016) based on market surveys. They are dependent on the nominal power of the equipment. They yielded cost estimates of 244 k€ per water turbine, and only 11 k€ for fresh water production, which is very small in comparison to the other costs.

Holl et al. also provide scaling laws for the cost of the electrolyzer and the vessel, based on the nominal power of the electrolyzer and the vessel length, respectively. Applying the scaling law to the 1,420 kW capacity electrolyzer of the energy ship results in an estimated cost of 1,845 k€, equivalent to 1,300 €/kW. This is greater than the 1,000 to 1,200 €/kW reported in (Gotz et al., 2016) and (Chardonnet et al., 2017) for alkaline electrolyzers (AEL). It is possible that Holl et al. may have considered PEM electrolyzers, which are more expensive (Gotz et al., 2016). Moreover, according to (Chardonnet et al., 2017), the cost of AEL electrolyzers is expected to decrease to 900 €/kW by 2025. Therefore, we used the range 900 to 1,200 €/kW to estimate the electrolyzer cost, yielding a final cost of 1,280 to 1,700 k€.

The scaling law that Holl et al. developed for vessel cost was based on data for sailing ships of length 10–20 m. Extrapolating from this data, an 80 m long vessel would cost in the order of 11 M€, equivalent to a specific cost of 42,000 € per tonne of structural mass, which is very high. In contrast, data from (Lindstad et al., 2012) and (Papanikolaou, 2014) suggest that the price of commercial ships is in the range 3,000 to 16,000 € per tonne of structural mass (steel), depending on the type and size of ship. However, lightweight materials such as aluminium or glass fibre reinforced polymers (GFRP) may be required to achieve sufficient structural strength in the structural mass budget of the energy ship. The greater cost of those materials (approx. 1 €/kg for steel, 3 €/kg for aluminium, 6 €/kg for GFRP) would increase the specific cost to 5,000 € per tonne for aluminium or 8,000 € per tonne for GFRP. As the hull material is unknown at this stage, we retain a cost range of 5,000 to 20,000 € per tonne, corresponding to a hull cost in the range 1,290 to 5,160 k€.

According to (Kuuskoski, 2019), the cost of four Flettner rotors is in the range 3,000 to 3,500 k€.

According to (Anicic et al., 2014), the cost of a power-to-methanol plant is approximately 81% of the cost of the electrolyzer, yielding a cost of 1,030 – 1,380 k€ for the energy ship's hydrogen-to-methanol plant. For the liquid $CO_2$ and methanol storage tanks, suppliers and prices can be found on the internet (e.g. https://www.gitank.com/methanol-storage-tanks, (Chart, 2019)); typical costs are 300 €/tonne of capacity for methanol and 1,000 €/tonne of capacity for liquid $CO_2$.



Finally, ship common systems, ship assembly and systems integration represent 20% of the total cost according to (Shetelig, 2013). Applying this ratio to the energy ship leads to an estimated total capital cost of approximately 8,570 to 14,700 k€ for

a first of a kind energy ship.

## 3.2 Capital cost of a first of a kind FARWIND energy system

According to the discussion in section 2.3, a FARWIND energy system should include a fleet of 112 energy ships and 5 tankers. One can expect the unit cost for a fleet of 112 energy ships to be significantly smaller than the cost of an energy ship protoype. To take this into account, a learning rate of 10% was assumed on the unit cost of the energy ship as function of the

built capacity, see Tab. 2. It can be noted that such learning rate corresponds to what was observed for wind turbines (Lindman and Soderholm, 2012). It leads to a range of capital cost of 545 to 938 M€ for the first fleet of energy ships. It corresponds to an average unit cost of 4,550 to 7,830 k€ per energy ship.

For the tanker, according to (Lindstad et al., 2012), the price of commercial ships is in the range 500 € to 4,750 € per tonne of dwt, depending on the type of ships and size. The lower price is for crude oil tankers with greater than 140,000 dwt, while

the higher price is for roll-in/roll-off (ro-ro) ships of 7,000 dwt. In the present study, we retain a cost range of 2,500 to 4,000 €/tonne of deadweight, leading to a tanker cost in the range 15,800 to 25,300 k€.

Thus, overall, the total capital cost of a FARWIND system comprised of 112 energy ships and 5 tankers is expected to be in the range of 625 to 1,065 M€, for an approximately 200 MW capacity. For the sake of comparison, the capital cost of the 498 MW bottom-fixed offshore wind farm near Fécamp (France) is approximately 2 billion euros. Therefore, the capital

costs per installed megawatt are comparable (3,125 to 5,325 k€/MW for the FARWIND system vs 4,000 M€/MW for the Fécamp wind farm).

## 3.3 Operational expenditures

Expected operation and maintenance (O&M) costs, including the cost of $CO_2$ supply, are summarized in Tab. 3 and detailed below.


|  | O&M cost (in % of capital cost of equipment per year) |
|---|---|
| **Energy ship** |  |
| Hull | 1 - 2% |
| Flettner rotors | 7% |
| Water turbine | 4 - 13% |
| Auxiliaries | 7% |
| Electrolyzer | 4% |
| $H_2$-to-methanol plant | 4% |
| Fresh water production unit | 10 - 20% |



| | |
|---|---|
| Liquid $CO_2$ tank | 1 - 2% |
| Methanol tank | 1 - 2% |
| **Total** | **4.8 – 5.1%** |
| **Tanker** | 4 - 10% |
| **FARWIND system** | |
| Energy ships maintenance | 28 – 45 M€/y |
| Tankers O&M | 3.2 – 13 M€/y |
| $CO_2$ supply | 2.8 – 28 M€/y |
| **Total (including $CO_2$ supply)** | **5.4 – 8.1%** |

Table 3 Estimates of the operation and maintenance costs of the optimal FARWIND system

### 3.3.1 Energy ships and tankers operation and maintenance cost

According to (Holl et al., 2016), the maintenance cost of the water turbine is in the range 4 to 13% of the capital cost, and that of the fresh water production unit is between 10 and 20%. According to (Chardonnet et al., 2017), the maintenance cost for the electrolyzer is in the order of 4% of capital cost. The same ratio is assumed for the hydrogen to methanol plant.

For the Flettner rotors and the auxiliaries, it is assumed that the maintenance cost is similar to that of offshore wind turbines, which is 7% of capital cost according to (Moné et al., 2015). For the other subsystems (hull, storage tanks), it is expected that the maintenance costs would be small; a range of 1 to 2% maintenance cost was arbitrarily selected. Overall maintenance costs for the energy ship are thus in the order of 5%.

For the tanker, following (Holl et al., 2016), we estimate operation and maintenance costs to be 4 to 10%.

### 3.3.2 $CO_2$ supply cost

The ambition of the FARWIND energy system is to provide a sustainable alternative to the use of liquid fossil fuels (e.g. oil). Therefore, as mentioned in the introduction, the $CO_2$ must be captured directly or indirectly from the atmosphere.

According to (Keith et al., 2018), the cost for direct air capture (DAC) using large-scale wet absorption DAC technology is in the range 80 to 204 €/tonne of $CO_2$. The cost of $CO_2$ capture from biogas upgrading is in the order of 15 to 100 €/tonne of $CO_2$ (Li et al., 2017). In the case of $CO_2$ capture from flue gases from combustion of biomass or FARWIND-produced methanol, the cost of carbon capture is in the order of 35 to 50 €/tonne (assuming that it would be similar to that for capture of $CO_2$ from power production processes involving coal or natural gas (Irlam, 2017)). Note that for both biogas upgrading and biomass or methanol combustion, the $CO_2$ concentration in the source is much greater than in ambient air, which results in a more effective capture than with DAC.

Carbon dioxide may also be extracted from seawater (Willauer et al., 2012). Indeed, some of the $CO_2$ present in the atmosphere dissolves in the ocean. However, this new technology is in its early stages of development (Willauer et al., 2017).



In any case, the captured $CO_2$ must be liquefied for efficient transportation. The energy requirement for $CO_2$ liquefaction is

in the order of 0.1 $kWh/kg_{CO2}$ according to (Oi et al., 2016), which is low; hence its associated cost is expected to be negligible.

Therefore, we estimate the cost of $CO_2$ production to be in the range 20 to 200 €/tonne. As approximately 140,000 t of $CO_2$ are required to produce 100,000 t of methanol, the $CO_2$ supply cost is estimated to be in the range 2,800–28,000 k€ per annum.

**4 Cost of energy estimates**

**4.1 Short-term cost**

The levelized cost of methanol *LCOM* can be calculated as (Holl et al., 2016):

$$LCOM = \frac{CRF + \lambda}{AMP} I$$

(1)

where $I$ is the total capital cost, $\lambda$ is the total O&M rate, $AMP$ is the annual methanol production, and $CRF = \frac{i(1+i)^n}{((1+i)^n - 1)}$ is the

capital recovery factor, in which $i$ is the interest rate on capital and $n$ is the lifetime in years.

Assuming an interest rate in the range 6–10% and a lifetime of 20 years, the capital recovery factor is in the range 8.7–11.7%. The methanol cost is thus in the range 0.87–2.08 €/kg (157 to 376 €/MWh$_{th}$).

This cost is two to five times greater than current market price for methanol (360 €/t ≈ 65 €/MWh in the first quarter of 2019). However, it does not take into account a price on GHG emissions. At least 0.675 kg of $CO_2$ is produced per kg of

methanol produced using conventional processes (which are based on coal or natural gas) (Martin and Grossmann, 2017). In 2018, the carbon tax was 44.6 €/tonne in France and 110 €/tonne in Sweden; if $CO_2$ emissions were taken into account in these countries, the methanol price would increase by 6 €/MWh$_{th}$ and 13 €/MWh$_{th}$ respectively. Thus, unfortunately, even with a rather significant carbon tax, the cost of FARWIND-produced methanol would not be competitive in the short term.



## 4.2 Long-term cost

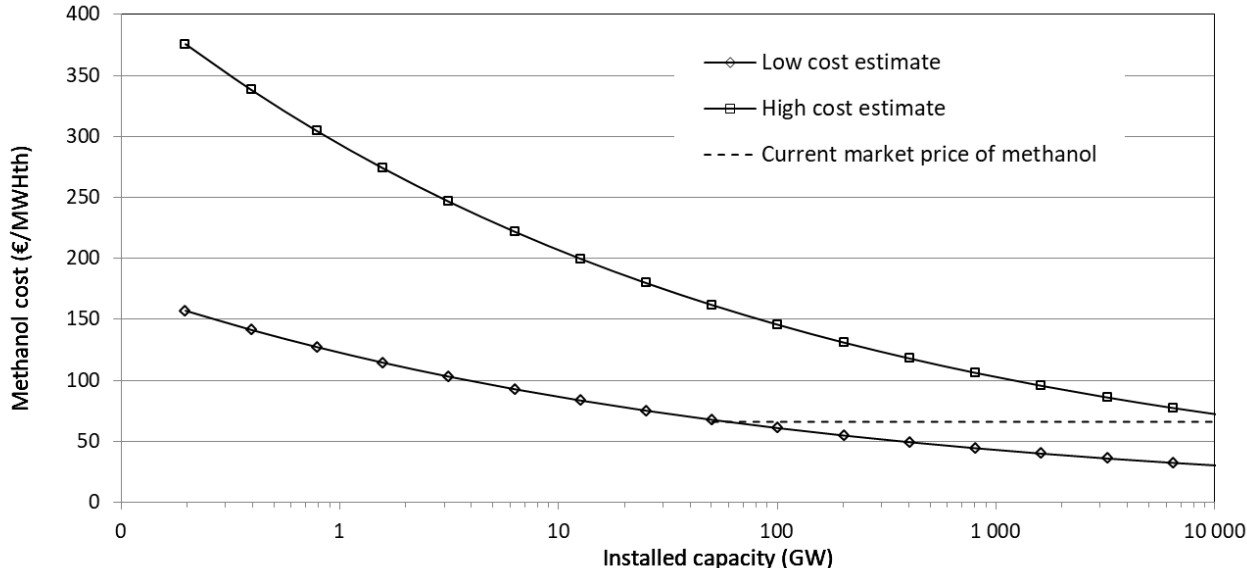


**Figure 3 Cost of methanol produced by FARWIND systems as function of the installed capacity and comparison with the current market price of methanol produced from fossil fuels and feedstocks**

However, as for the energy ships, one can expect that the cost of FARWIND systems will decrease with increasing installed capacity. Fig. 3 shows the expected cost reduction for the methanol cost as function of the installed capacity. A learning rate

of 10% was assumed (as for the energy ships, see section 3.2). One can see that it would take hundreds to thousands of GW of installed capacity to achieve competitiveness with methanol produced from fossil fuels.

## 4.3 Market potential

Finally, let us consider the perspective of FARWIND-produced fuel for the transportation fuel market. Indeed, methanol can be blended with gasoline in low quantities for use in existing road vehicles. According to (Methanol Institute, 2014), the

blend can include up to 15% methanol by volume (M15 fuel). Moreover, flexible fuel vehicles which can run on an 85%–15% methanol–gasoline mix (M85 fuel) have been developed and commercialized (e.g. the 1996 Ford Taurus); and M100 (100% methanol) vehicles are in development (Olah et al., 2018). Thus, FARWIND-produced methanol could be used as a low-carbon substitute to oil on the transportation fuel market.



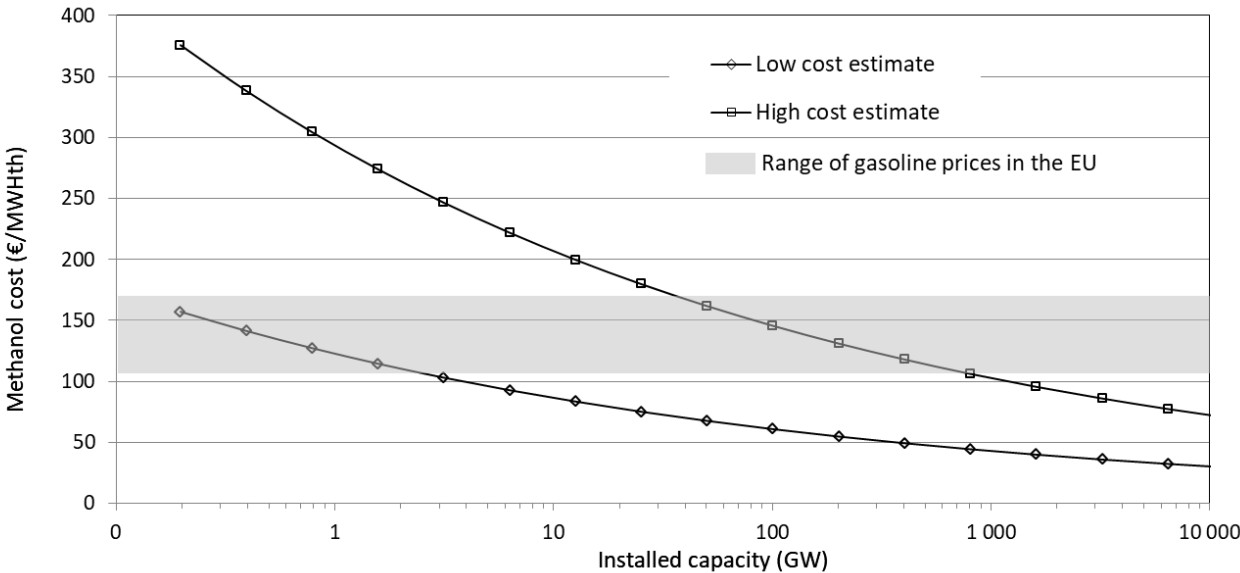

**195**    **Figure 4 Cost of methanol produced by FARWIND systems as function of the installed capacity and comparison with current market price of gasoline in the EU**

Let us compare the cost of FARWIND-produced methanol to the gasoline price in the EU. Gasoline price ranges from 1.1 €/L (Bulgaria) to 1.7 €/L (Netherlands), the price differences arising from different policies on fuel taxes in different countries (European Commission, 2019). This price range is equivalent to 112 to 173 €/MWh$_{th}$, since the standard density of

**200**    gasoline traded in the EU is 0.755 kg/L and its energy content is approximately 13 kWh$_{th}$/kg. Thus, as can be seen in Fig. 4 and provided that taxes policies are favourable to FARWIND-produced methanol, it could take "only" a few GW to a few tens of GW of installed capacity to be competitive with gasoline on the EU transportation fuel market.

## 5 Conclusions

In this paper, we presented a concept for sustainable methanol production from the far-offshore wind energy resource. It is

**205**    based on autonomous fleets of energy ships for the fuel production and manned tankers for the collection and transport of the produced methanol, as well as the supply of $CO_2$ to the energy ships

The proposed FARWIND energy system includes a fleet of 112 energy ships supported by five dedicated tankers. Its methanol production is expected to be in the order of 100,000 t per annum (approximately 550 GWh per annum of chemical energy). The cost of this methanol is expected to be in the range 0.87–2.08 €/kg for the first-of-a-kind FARWIND system,

**210**    which is significantly greater than the current market price for fossil fuel-derived methanol (0.36 €/kg). However, methanol can be used as a substitute to fossil fuels on the fuel transportation market: since the price of transportation fuel is high in most European countries, and assuming that a cost reduction similar to that observed for land-based wind energy can be achieved, the cost of FARWIND-produced methanol could compete with gasoline in the EU.



Nevertheless, it must be acknowledged that the FARWIND system is only in the early stages of development. Future work is
required to confirm the potential: challenges include the development and validation of the key subsystems (water turbine, autonomous power-to-methanol plant, control systems for autonomous navigation) and addressing the possible non-technical barriers to far-offshore wind energy (legal status of autonomous far-offshore wind energy converters, environmental impacts).

## 6 Acknowledgements

This research was partially supported by the French National Energy and Environmental Agency (ADEME) and Région Pays de la Loire.

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
