# Peer review of "Exploitation of the far-offshore wind energy resource by fleets of energy ships. Part B. Cost of energy"

_Wind Energy Science, 2019_

## Referee Comment (RC1) · Anonymous Referee #1 · 24 Feb 2020

This paper presents a new concept for converting far-offshore wind energy into methanol. The idea to utilize the rich far-offshore wind resources to produce fuel is compelling. The choice of producing methanol instead of a $CO_2$-free fuel like $H_2$ or $NH_3$ seems a bit arbitrary, and should be better argued. A sub-section should be included to compare alternative fuels that could be produced offshore.

The paper estimates the cost of producing methanol. It specifies the assumed cost elements, but does not show clearly how much the various cost elements contribute to the calculated total cost for producing methanol from energy ships. The paper should include a graph that clearly show the contribution of each cost element to the total cost.

[Figure]

It should also be discussed if the assumed cost elements are realistic. For example, it is stated that the assumed cost pr MW installed generation capacity is about the same as for a bottom fixed wind farm (4000 EUR/MW). To judge if this is a fair estimate or not it would be good to compare the two constructions in terms of complexity, amount of steel, etc.

The calculated cost for methanol from energy ships is 0.87–2.08 €kg (157 to 376 €MWh_th). This is according to the paper 2 to 5 times the current market price for methanol, but can be competitive if cost reduction is achieved. It is stated that a learning rate of 10 % would make the system cost competitive if installing hundred or thousands of GW of methanol producing energy ships, or just a few GW if competing against gasoline price in EU. These are all relevant comparisons, but would be more interesting if the comparison could be made in more detail: this should include showing how much the various cost elements contribute to the calculated total cost for producing methanol from energy ships, and also discuss if the same learning rate can be applied to all the cost elements.

This should be discussed: What is the cost per kWh_el from the water turbine, and what is the cost of the power to methanol plant? If the cost of electricity from the water turbine in this concept is overly expensive, would it not make more sense to produce methanol on plant connect to a more standard wind farm onshore or offshore?

---

## Author Comment (AC1) · 28 Feb 2020

Dear anonymous referee #1.

Thanks for having read our paper and thanks for your comments. Please find below our answers and clarifications.

1. Choice of methanol rather than CO2-free fuels. We have already published a comparison of the various options in (Babarit et al., 2019) and it is summarized in the introduction of the companion paper of this paper: https://www.wind-energ-sci-discuss.net/wes-2019-100/ Nevertheless, we will add the following text in the intro-

[Figure]

duction of the revised version of the paper:

"In the proposed system, the fuel is methanol (see Babarit et al., submitted and Babarit et al., 2019 for detailed explanation of the choice of methanol rather than CO2-free fuels like hydrogen or ammonia)."

2. Contribution of each cost element to total cost. We will add a figure (see below) and the following discussion in section 4.1 in the paper's revision.

"Figure 3 shows the cost breakdown for an average cost scenario. One can see that the main cost sources are the financing cost (30% of total methanol cost), the FAR-WINDERs' capital cost (ship + power-to-methanol plant + integration, 25% of total methanol cost), and operation and maintenance cost of the FARWINDERs (24%). The total cost of energy storage - including the power-to-methanol plants, CO2 supply and tankers capital cost, and operation and maintenance cost - accounts for 25% of total cost."

3. Credibility of cost elements. We put a lot of effort in looking for reliable cost sources. It included consultation with possible suppliers and industry representatives. The cost data which is used in the paper is the best that we have been able to gather. Moreover, we included cost ranges in order to reflect uncertainties in some of the cost data. Therefore, we believe that the cost elements and the calculated range for the methanol cost are realistic. Nevertheless, note that according to IEA Wind TCP Task 37 Technical report "systems engineering in wind energy: WP2.1 reference wind turbines", May 2019, the total mass of a 3.4 MW land-based wind turbine is 820 tons and its initial capital cost is in the order of 3,800 k€It corresponds to a 4.6 € / kg cost to mass ratio (respectively 1 120 € / kW cost to capacity ratio) whereas the range of cost to mass ratio in the paper is 13.0 – 22.3 € / kg for a FARWINDER prototype (4 800 - 8100) and 7.4 – 12.7 € / kg for the 112 FARWINDERs of the FARWIND system (2 700 - 4 650). As one can expect the cost to mass ratio (cost to capacity) is significantly greater for the FARWINDERs than for the wind turbine which can be explained by a greater

complexity.

4. Different learning rates depending on the cost elements. We agree that this refinement would be interesting, but we believe that it goes beyond the scope of the present paper which aims at providing first estimates for the medium and long term cost of energy.

5. Cost of kWh_el from the water turbine. The cost of power available to the power-to-methanol can be estimated to 44 – 99 € / MWh for the first FARWIND system, which is comparable to the cost of electricty from a conventional onshore or offshore wind farm. However, low cost methanol production requires both low cost of electricity and high capacity factor. This is discussed in the second paragraph of the introduction in the companion paper of this paper https://www.wind-energ-sci-discuss.net/wes-2019-100/ : "However, the main challenge faced by PtX products from renewable energy-based plants is cost competitiveness. Key economic drivers are the cost of input electricity to the PtX plant and the PtX plant capacity factor (Fasihi et al., 2016; Ioannou and Brennan, 2019)." The following text will also be added in section 4.2 in the paper's revision:

"As shown in (Fasihi et al., 2016; Ioannou and Brennan, 2019), the key economic drivers in power-to-gas or power-to-liquid processes are the cost of input electricity to the power-to-gas/liquid plant and the power-to-gas/liquid plant capacity factor. For FARWIND systems, the cost of input electricity LCOE can be estimated by using equation (1) but without taking into account costs related to energy storage (power-to-methanol plant, tankers, $CO_2$ supply). It is found to be in the range 44 – 99 € / MWh for a first-of-a-kind FARWIND system, which is comparable to the cost of electricity from land-based and bottom-fixed offshore wind turbines. However, as the capacity factor of FARWIND systems is expected to be significantly greater, the cost of methanol produced by FARWIND systems is expected to be cheaper."

[Figure]

**Fig. 1.** Cost breakdown of methanol produced by FARWIND systems. The shown data corresponds to an average cost scenario (methanol cost equal to 1.48 €kg).

---

## Referee Comment (RC2) · Anonymous Referee #1 · 2 Mar 2020

Thank you for your good answers to my comments. These were clarifying for me. It is good that you will include a pie-chart to show the various cost elements for producing the fuel. Is the stated cost of finance calculated as the assumed interest rate of the investments? And which interest rate and lifetime was assumed?

---

## Author Comment (AC3) · 2 Mar 2020

We assumed a lifetome of 20 years and an interest rate of 6% for the low cost scenario and 10% for the high cost scenario, which gives a capital recovery factor in the range 8.7-11.7%. This information can be found in section 4.1: "Assuming an interest rate in the range 6–10% and a lifetime of 20 years, the capital recovery factor is in the range 8.7–11.7%."
* * *

---

## Referee Comment (RC3) · Anonymous Referee #2 · 30 Apr 2020

[referee-annotated manuscript omitted]

---

## Author Comment (AC4) · 1 May 2020

Thank you for your comments. Indeed the technical viability and design is discussed in the first paper. The typos have been corrected, thanks for pointing them out.